# Heart Rate Variability and Its Associations with Organ Complications in Adults after Fontan Operation

**DOI:** 10.3390/jcm10194492

**Published:** 2021-09-29

**Authors:** Magdalena Okólska, Jacek Łach, Paweł T. Matusik, Jacek Pająk, Tomasz Mroczek, Piotr Podolec, Lidia Tomkiewicz-Pająk

**Affiliations:** 1Cardiological Outpatient Clinic, Department of Cardiovascular Diseases, John Paul II Hospital, 31-202 Krakow, Poland; m.okolska@interia.pl; 2Department of Cardiac and Vascular Diseases, Institute of Cardiology, Jagiellonian University Medical College, John Paul II Hospital, 31-202 Krakow, Poland; djholter@interia.pl (J.Ł.); ppodolec@interia.pl (P.P.); ltom@wp.pl (L.T.-P.); 3Department of Electrocardiology, Institute of Cardiology, Faculty of Medicine, Jagiellonian University Medical College, John Paul II Hospital, 31-202 Krakow, Poland; 4Department of Pediatric Heart Surgery and General Pediatric Surgery, Medical University of Warsaw, 02-091 Warsaw, Poland; jacekpajak@poczta.onet.pl; 5Department of Pediatric Cardiac Surgery, Jagiellonian University, 30-663 Krakow, Poland; t_mroczek@hotmail.com

**Keywords:** heart rate variability, physical performance, Fontan operation, organ complications

## Abstract

Reduction of heart rate variability (HRV) parameters may be a risk factor and precede the occurrence of arrhythmias or the development of heart failure and complications in people with postinfarct left ventricular dysfunction and after coronary artery bypass grafting. Data on this issue in adults after a Fontan operation (FO) are scarce. This study assessed the association between HRV, exercise capacity, and multiorgan complications in adults after FO. Data were obtained from 30 FO patients (mean age 24 ± 5.4 years) and 30 healthy controls matched for age and sex. HRV was investigated in all patients by clinical examination, laboratory tests, echocardiography, a cardiopulmonary exercise test, and 24-h electrocardiogram. The HRV parameters were reduced in the FO group. Reduced HRV parameters were associated with patients’ age at the time of FO, time since surgery, impaired exercise capacity, chronotropic incompetence parameters, and multiorgan complications. Univariate analysis showed that saturated O_2_ at rest, percentage difference between adjacent NN intervals of >50 ms duration, and peak heart rate were associated with chronotropic index. Multivariable analysis revealed that all three variables were independent predictors of the chronotropic index. The results of this study suggest novel pathophysiological mechanisms that link HRV, physical performance, and organ damage in patients after FO.

## 1. Introduction

The Fontan operation (FO) is the treatment of choice for patients with single-ventricle congenital heart disease [1]. The aim of this operation is to separate the pulmonary and systemic circulation, and achieve normal or near-normal arterial oxygen saturation. However, over time, cardiac and extra-cardiac complications develop in patients after FO. The literature also provides reports on the development of pathophysiological abnormalities, including abnormal functioning of the autonomic nervous system, altered chemoreceptor activity, and neurohumoral disorders, in this patient population [2,3,4].

One of the most significant problems in clinical practice during follow-up is identifying patients with a high risk of mortality. Parallel to the use of well-known methods (such as echocardiography), new methods are being developed, but their promising diagnostic or prognostic value is still not fully understood. Assessment of heart rate variability (HRV) is one such method.

A severe decrease in HRV indicates autonomic nervous system (ANS) dysfunction [5]. Decreased HRV may be a risk factor and precede organ complications. Research on patients with postinfarct left ventricular dysfunction indicates a relationship between reduced HRV and the development of heart failure, a higher risk of ventricular arrhythmia, and a worse prognosis [6,7]. HRV reduction is also observed in patients after cardiac surgery [8]. In adults with congenital heart disease, reduced HRV parameters were found among those with tetralogy of Fallot, systemic right ventricle, or cyanotic heart disease and those who underwent aortic coarctation repair or ventricular septal defect closure with right bundle branch block [9,10,11,12]. However, only a few studies have analyzed these aspects in patients after FO, and thus the data are limited. Furthermore, literature reports are mostly related to a pediatric population [13,14,15,16,17,18], while data regarding adult patients are scarce.

Therefore, this study aimed to assess the relationship between HRV parameters, exercise capacity, and multiorgan complications in adults after FO.

## 2. Materials and Methods

### 2.1. Study Participants

This was a retrospective study and included 30 adult patients over 18 years of age. All the patients underwent FO as they were diagnosed with functionally single ventricular heart. The patients remained under medical supervision in John Paul II Hospital. The exclusion criteria of the study were as follows: diagnosis of pulmonary artery hypertension requiring vasodilator therapy, asthma, atrial flutter, atrial fibrillation, diabetes, current infection, inflammation, and neoplastic disease, major trauma, pregnancy, use of vitamin K antagonists or beta-blockers, and history of pacemaker placement and alcohol abuse. Healthy, age- and sex-matched volunteers were included in the control group.

All the demographic, anatomic, and clinical data required for the study were obtained from the patients’ medical records. Each patient was subjected to a physical examination as well as an assessment of body mass index, ejection fraction of the systemic ventricle, and arterial oxygen saturation. Body mass index was calculated by dividing the weight of the patient (kg) by height (m^2^). Oxygen saturation was measured by pulse oximetry while breathing room air.

### 2.2. Echocardiography

Ejection fraction of the systemic ventricle was assessed using Simpson’s method. In addition, valvular competence was evaluated in all the patients by two experienced, independent cardiologists using echocardiography (Vivid 7, GE Medical Systems, Milwaukee, WI, USA), as previously described [19].

### 2.3. Laboratory Investigations

After overnight fasting for at least 12 h, blood samples were collected from the antecubital vein of patients. The samples were evaluated for the following laboratory parameters that may indicate multi-organ complications: white blood cell count, red blood cell count, hemoglobin concentration, hematocrit, red blood cell distribution width, platelet count, mean platelet volume (MPV), total protein, alanine aminotransferase, aspartate transaminase, gamma-glutamyl transpeptidase (GGTP), alkaline phosphatase, total bilirubin α-fetoprotein, creatinine, cystatin C and N-terminal pro-B-type natriuretic peptide. All of them were assessed by routine laboratory techniques.

### 2.4. Cardiopulmonary Exercise Test

Exercise tolerance was determined by performing the cardiopulmonary exercise test (CPET) using a modified Bruce protocol (Reynols Medical System, ZAN-600, Hertford, UK). The following parameters were recorded during CPET: blood pressure, rest oxygen saturation (Sat. O_2_ rest), 12-lead electrocardiogram, time of exercise, minute ventilation (VE), peak oxygen uptake (VO_2_ peak), respiratory exchange ratio (RER), peak ventilatory equivalent of oxygen (VE/VO_2_), peak ventilatory equivalent of carbon dioxide (VE/VCO_2_), and breathing reserve. VO_2_ peak was estimated as the highest oxygen uptake at peak exercise (mL/kg/min), and the percentage of the predicted value was calculated. Ventilatory anaerobic threshold was measured using the V-slope method. VE/VO_2_ was defined as the amount of ventilation needed to uptake a given amount of oxygen, while VE/VCO_2_ was defined as the amount of ventilation needed to eliminate a given amount of carbon dioxide. RER was calculated by dividing VO_2_ by VCO_2_.

### 2.5. Chronotropic Incompetence

Chronotropic index was determined based on the chronotropic metabolic relationship introduced by Wilkoff et al. [20] and calculated by using the following formula: (peak heart rate − resting heart rate)/(220 − age − resting heart rate). Chronotropic incompetence was defined as a chronotropic index value of <0.8.

Heart rate reserve (HRR) was calculated as the difference between maximal heart rate (HRmax) and peak heart rate. HRmax was determined using the following formula: 220 − age. Accordingly, HRR was calculated as follows: HRR = HRmax − peak heart rate = 220 − age − peak heart rate [21,22].

### 2.6. Ambulatory 24-h Holter Electrocardiogram

All patients and controls were subjected to standard 24-h electrocardiographic monitoring during daily activity, using a commercially available Holter system. All Holters were reviewed by two experienced observers. All recordings were analyzed using a PC-based Holter system, and those shorter than 21 h were excluded. The predominant rhythm was defined as the one that was present during >50% of the time during the Holter recording.

### 2.7. Heart Rate Variability

All Holters with available data were reviewed by two experienced analysts to analyze HRV. The beats were classified by automated software as normal, supraventricular extrasystolic, ventricular extrasystolic, those of uncertain origin, or artifacts. The classification was manually reviewed and corrected if necessary. Only normal-to-normal (NN) intervals were included in the HRV analysis.

The following HRV time-domain parameters were measured: standard deviation of all NN intervals (SDNN), standard deviation of the averages of NN intervals in all 5-min segments of the entire recording (SDANN), root mean square of the differences of successive NN intervals (rMSSD), percentage difference between adjacent NN intervals of >50 ms duration (pNN50), and HRV triangular index.

Furthermore, the following HRV frequency parameters were measured: very low frequency (0.017–0.050 Hz); low frequency (0.050–0.150 Hz); and high frequency (0.150–0.350 Hz).

The high frequency component is associated with the activity of the parasympathetic nervous system, and low frequency with both the sympathetic and parasympathetic system. Very low frequency is related to more long-term fluctuations in heart rate [23].

The total power was determined at frequencies ranging from 0.017 to 0.050 Hz. In addition, the low frequency/high frequency ratio was calculated. All spectral indexes were calculated as average data over the complete recording period (up to 24 h). The parameters SDNN, HRV triangular index, total power, and low frequency were assumed to reflect, with some simplification, the overall HRV or activity of both sympathetic and parasympathetic components of the ANS, while the parameters rMSSD, pNN50, and high frequency were directly proportional to the parasympathetic nervous system [23].

### 2.8. Statistical Analysis

The data distribution was presented as numbers and percentages for categorical variables, means with SDs for normally distributed continuous variables, and medians with lower and upper quartiles (Q1–Q3) for continuous variables with non-normal distribution. The normality of the data distribution was verified using the Kolmogorov–Smirnov test. Quantitative variables of patients who underwent FO and control participants were compared using the two-tailed Student’s *t*-test or Mann–Whitney *U* test, whereas qualitative variables were analyzed using the chi-square test. The association between numerical variables was analyzed by calculating the Pearson’s correlation or Spearman’s rank correlation coefficient. Moreover, the simultaneous influence of Sat. O_2_ rest and pNN50 on peak heart rate was assessed using the linear regression model. The results were presented as coefficient (*b*) with 95% confidence interval. The *R*-square value was calculated to describe the goodness-of-fit for the linear regression model. All the analyses were performed using IBM SPSS Statistics for Windows, Version 25.0 (IBM Corp., Armonk, NY, USA). Statistical significance was defined as *p* < 0.05 for the two-tailed test.

## 3. Results

### 3.1. Patients’ Characteristics

Thirty adult patients who underwent FO were enrolled in the study, including 17 men (57%) with a mean age of 24 ± 5.4 years. These patients did not show any significant difference from the controls with regard to age, sex, and body mass index. The median age of patients at the time of surgery was 3 (Q1–Q3: 2–5) years, the median time after surgery was 19.5 (Q1–Q3: 17–21) years. A total of 19 (64%) patients had fenestration, and 11 (36%) had no fenestration. The mean ejection fraction of the systemic ventricle was 52 ± 9.1%. The baseline characteristics of the study group and the control group are presented in Table 1 and Table 2.

### 3.2. Laboratory Tests Results

The laboratory parameters determined for the FO group and the control group are presented in Table 3.

### 3.3. CPET Results

The CPET results of patients from the FO group were compared with those from the control group and are presented in Table 4.

### 3.4. Heart Rate Variability

A significant reduction in HRV was observed in adult patients with Fontan circulation compared with the control subjects (Table 5).

The correlations between the HRV parameters and patients’ characteristics are presented in Table 6.

### 3.5. Relationship between HRV, CPET, and Chronotropic Incompetence Parameters

The results of univariate analysis showed that Sat. O_2_ rest, pNN50, and peak heart rate were associated with chronotropic index (the strongest relationship was observed for peak heart rate (R^2^ = 0.54)). The results of multivariable analysis showed that all three variables were significant predictors of chronotropic index, accounting for 70% variability in chronotropic incompetence (Table 7).

## 4. Discussion

This study assessed the association between HRV, exercise capacity, and multi-organ complications in adults after undergoing FO. The findings revealed that patients after FO had significantly reduced HRV, implying a correlation between HRV parameters and age at the time of surgical intervention, the time since operation, reduced exercise capacity, and organ complications.

The activity of the ANS regulates HRV measures. In this study, 70% of patients from the study group showed a significant reduction in HRV parameters. These observations are in accordance with the earlier studies that investigated this issue in pediatric patient groups, but studies on adult groups are scarce [2,14,15,17,24].

In our study, adult patients who underwent FO showed a decrease in HRV parameters. This shows the association between age and reduced HRV. The reduction progressed over time after surgery. Similar observations were presented in the studies by Dahlaqvist et al. [15] and Rydberg et al. [17]. The age of the patient at the time of the surgery plays a crucial role in the pathogenesis of arrhythmias. According to the literature, the best time for surgery in children with single-ventricle heart is up to 4 years of age [25,26,27]. Decreased HRV parameters were shown in children regardless of the surgical intervention approach utilized (intra-atrial lateral tunnel or extra-cardiac conduit) [15]. The findings of our study may be considered valuable and suggest that special attention should be paid for early qualification of children to FO. This observation is consistent with previous reports, including that of Abbott et al. [28], who proved that an increased preoperative heart rate is associated with an increased perioperative risk of heart damage and mortality.

The CPET results of this study revealed that reduced HRV parameters were associated with chronotropic incompetence and exercise capacity.

Patients who had decreased HRV parameters had lower VO_2_ peak and percentage of predicted value of VO_2_ peak, and higher VE/VCO_2_.

It has been shown that patients with overt heart failure had decreased VO_2_ peak and increased VE/VCO_2_, which are considered well-established predictors of mortality [29,30,31]. Furthermore, Kyoto et al. [32] reported a correlation between increased heart rate and decreased VO_2_ peak, independent of age, sex, and heart disease. Silvilaired et al. [33] examined patients after tetralogy of Fallot and observed a relationship between the HRV parameters (low frequency and high frequency) and reduced VO_2_ peak, which suggested that an impaired ANS response may be responsible for decreased exercise tolerance.

It has also been shown that VE/VCO_2_ reflects the relationship between minute ventilation and CO_2_ excretion [34]. This study revealed that patients with heart failure exhibited an elevated VE/VCO_2_ that was associated with excessive minute ventilation in relation to exercise effort. One of the possible mechanisms explaining this phenomenon is an imbalance between the ANS and excessive sympathetic activity. Previous studies conducted among people with heart failure have shown that an increased ventilation response, expressed as higher VE/VCO_2_, independently correlates with all HRV parameters [35].

Impaired chronotropic response is another issue observed in patients who have had FO [30,36]. Patients after FO have higher HRR, achieve shorter exercise time in CPET, and show lower VO_2_ peak and higher VE/VCO_2_ values. In this study, we attempted to explore whether the knowledge on reduced HRV parameters might be used to predict the occurrence of organ complications. This assumption seems to be justified: our results confirmed that patients with lower pNN50, Sat. O_2_ rest, and peak heart rate had chronotropic incompetence and 70% probability of developing heart failure.

In patients after Fontan surgery, liver disorders commonly occur and may cause serious clinical complications [37,38]. In this study, we registered an increased level of GGTP and showed a correlation between GGTP and pNN50. This increased level of GGTP observed in patients after FO could be attributed to liver dysfunction or damage caused by chronic blood stagnation due to increased venous pressure (prevailing in Fontan’s circulation). To our knowledge, no studies have so far analyzed the relationship between reduced HRV parameters and liver dysfunction in patients after FO. However, several studies in the literature indicate reduced HRV values in patients with liver fibrosis [39,40,41]. These studies suggest that ANS dysfunction is associated with poor prognosis in this patient population. Furthermore, Bohogal et al. [41] showed that specific HRV parameters, regardless of the Model for End-Stage Liver Disease score, can predict mortality in patients with cirrhosis. The mechanism of ANS dysfunction in patients with liver fibrosis is still unknown and requires further research. However, the association between HRV and GGTP as observed in our study may be suspected to indirectly indicate the risk of developing liver dysfunction.

Holter ECG monitoring is relatively simple, generally available and non-invasive. It is performed in every patient after FO. HRV parameters have been identified as useful in various clinical scenarios [5,42]. We suppose that decreased HRV parameters may help to identify patients with specific organ complications. Decreased HRV parameters may be a risk of future arrhythmias and may indicate the need for regular follow-up in the case of heart failure development.

This retrospective study has several limitations to be acknowledged. Firstly, the number of patients was small and relatively mixed. Secondly, HRV parameters are highly sensitive to external factors. The norms for the analysis of HRV parameters and prognostically significant reduced values of these parameters in patients with congenital heart disease have not yet been developed. Therefore, further analyses with larger groups of patients are necessary.

## 5. Conclusions

This study revealed that patients after FO had reduced HRV parameters indicating ANS dysfunction, which was found to be associated with lower exercise tolerance and poor liver function. The data of the study suggest novel pathophysiological mechanisms that link HRV, physical performance, and organ damage in patients after FO.

## Figures and Tables

**Table 1 jcm-10-04492-t001:** Baseline characteristics of the study group and controls.

Variables	Fontan Patients (*n* = 30)	Controls (*n* = 30)	*p*-Value
Age, years	24 (5.4)	25.6 (3.8)	0.23
Female sex, *n* (%)	13 (43)	12 (40)	0.95
Height, cm	170 (8.1)	173 (6.9)	0.19
Weight, kg	65.08 (9.7)	69.0 (9.3)	0.85
Body mass index, kg/m^2^	22.5 (2.7)	22.7 (2.2)	0.69

Continuous data are presented as mean (SD) and categorical data as number (percentage).

**Table 2 jcm-10-04492-t002:** Baseline characteristics of patients after the Fontan operation.

Variables	Patients (*n* = 30)
Anatomic diagnosis, *n* (%)	
Tricuspid atresia	5 (17)
Pulmonary stenosis/TGA	4 (13)
Right ventricular hypoplasia	11 (36)
Hypoplastic left heart syndrome	5 (17)
Double-outlet right ventricle with left ventricular hypoplasia	3 (10)
Double-inflow left ventricle	1 (4)
Common atrioventricular canal	1 (4)
Systemic ventricle type, *n* (%)	
Left ventricle	24 (80)
Right ventricle	6 (20)
NYHA functional class, *n* (%)	
I	5 (17)
II	21 (71)
III	4 (12)
IV	0 (0)
Types of Fontan operation, *n* (%)	
Total cavopulmonary connection, lateral tunnel	29 (96)
Atriopulmonary connection	1 (4)

Abbreviation: NYHA, New York Heart Association; TGA, transposition of great arteries.

**Table 3 jcm-10-04492-t003:** Laboratory parameters in patients after Fontan procedure and in controls.

Variables	Fontan Group (*n* = 30)	Controls (*n* = 30)	*p*-Value
NT-proBNP, pg/mL	148.0 (96.0–470.0)	24.5 (6.0–35.0)	<0.001
RBC, 10^9^/μL	5.5 (0.6)	4.9 (0.5)	<0.001
Hemoglobin, g/dL	18.8 (1.8)	14.7 (1.3)	0.011
Hematocrit, %	47.6 (4.5)	43.0 (3.3)	<0.001
RDW, %	13.2 (12.9–14.4)	12.4 (12.0–12.6)	<0.001
Platelet count, 10^3^/μL	164.4 (70.8)	228.2 (38.1)	<0.001
PDW, fL	16.0 (3.2)	12.2 (2.3)	<0.001
MPV, fL	12.0 (1.2)	10.4 (1.0)	<0.001
Cystatin C, mg/L	0.9 (0.2)	0.8 (0.1)	0.009
Creatinine, μmol/L	72.8 (12.9)	77.4 (14.2)	0.19
eGFR, mL/min/1.73 m^2^	116.5 (13.6)	112.0 (12.9)	0.26
AST, IU/L	24.0 (20.0–28.0)	19.5 (17.0–22.0)	<0.001
ALT, IU/L	24.0 (19.0–27.0)	20.0 (17.0–23.0)	0.04
GGTP, U/L	61.5 (44.0–117.0)	15.5 (14.0–18.0)	<0.001
Bilirubin, μmol/L	18.3 (10.7–34.0)	12.0 (7.7–17.0)	0.002
α-Fetoprotein, ng/mL	2.5 (1.9–3.6)	2.3 (1.9–3.4)	0.657
ALP, U/L	80.5 (64.0–88.0)	67.0 (55.0–89.0)	0.11
Total protein, g/dL	75.1 (70.2–78.8)	75.0 (73.0–78.6)	0.43
Prothrombin time, s	13.6 (12.6–15.2)	11.9 (11.4–12.0)	<0.001
INR	1.2 (1.2–1.65)	1.0 (0.9–1.1)	<0.001
AST/ALT ratio	1.1 (0.4)	1.0 (0.3)	0.21

Continuous data are presented as mean (SD) or median (Q1–Q3). Abbreviations: ALP, alkaline phosphatase; ALT, alanine aminotransferase; AST, aspartate aminotransferase; AST/ALT ratio, ratio of aspartate transaminase to alanine transaminase; eGFR, estimated glomerular filtration rate; GGTP, γ-glutamyl transpeptidase; INR, international normalized ratio; MPV, mean platelet volume; NT-proBNP, N-terminal pro-B-type natriuretic peptide; PDW, platelet distribution width; RBC, red blood cells; RDW, red cell distribution width.

**Table 4 jcm-10-04492-t004:** Cardiopulmonary exercise test results of patients in the Fontan operation group and controls.

Variables	Fontan Group (*n* = 30)	Controls (*n* = 30)	*p*-Value
Exercise time, min	13.5 (3.4)	16.65 (2.7)	<0.001
Sat. O_2_ rest, %	92.0 (89.0–93.0)	97.0 (96.0–98.0)	<0.001
Sat. O_2_ exercise, %	87.0 (84.0–89.0)	97.0 (96.0–97.0)	<0.001
Peak VO_2_ per kg, mL/kg/min	20.6 (18.2–23.2)	50.9 (46.5–54.1)	<0.001
Peak VO_2_, %*n*	55.0 (48.0–63.0)	97.0 (95.0–98.0)	<0.001
VE	46.0 (35.0–63.0)	123 (97–138)	<0.001
VE/VCO_2_, L/L	33.3 (3.9)	26.5 (2.9)	<0.001
RER peak	1.0 (0.08)	1.1 (0.9)	0.01
Chronotropic index	0.55 (0.47–0.62)	0.93 (0.88–0.99)	<0.001
HRR	32.0 (24.0–60.0)	8.0 (1.0–14.0)	<0.001

Continuous data are presented as mean (SD) or median (Q1–Q3). Abbreviations: HRR, heart rate reserve; peak VO_2_ per kg, peak oxygen uptake per kilogram; peak VO_2_ (%*n*), percentage of predicted value for peak oxygen uptake; RER peak, peak respiratory exchange ratio; Sat. O_2_, oxygen saturation; VE, minute ventilation; VE/VCO_2_, peak ventilatory equivalent of CO_2_.

**Table 5 jcm-10-04492-t005:** Heart rate and HRV in the Fontan patients and control subjects.

Variables	Fontan Patients (*n* = 30)	Controls (*n* = 30)	*p*-Value
Heart rate, bpm	69.1 (10.4)	80.5 (6.5)	<0.001
Mean NN, ms	922.0 (157.9)	771.57 (59.4)	<0.001
SDNN, ms	121.8 (29.6)	152.74 (23.94)	<0.001
SDANN, ms	111.9 (31.6)	133.63 (25.2)	<0.001
rMSSD, ms	16.5 (10.9–33.5)	32.65 (27.4–43.7)	<0.001
pNN50, ms	6.75 (2.7–13.0)	11.8 (7.2–13.2)	0.018
HRV triangular index, ms	34.5 (11.3)	45.7 (7.8)	<0.001
Very low frequency (ms^2^)	301.6 (13.1–491.2)	491.8 (256.2–71.2)	0.030
Low frequency (ms^2^)	332.8 (93.4–551.2)	712.3 (538.3–1129.1)	<0.001
High frequency (ms^2^)	140.1 (46.1–303.2)	289.0 (156.7–370)	0.019
Total power (ms^2^)	861.1 (1.0–1738.0)	1618.7 (957.4–2031)	0.003
Low frequency/high frequency ratio	3.5 (2.5)	4.2 (1.5)	0.190

Continuous data are presented as mean (SD) or median (Q1–Q3). Abbreviations: HRV, heart rate variability; NN, normal-to-normal interval; pNN50, percentage difference between adjacent NN intervals of >50 ms duration; rMSSD, root mean square of the differences of successive NN intervals; SDANN, standard deviation of the averages of NN intervals in all 5-min segments of the entire recording; SDNN, standard deviation of all NN intervals.

**Table 6 jcm-10-04492-t006:** Correlations between HRV parameters and patient characteristics.

	Share of the Autonomous Components in the Modulation of HRV Parameters						
	Overall HRV or Sympathetic and Parasympathetic Nervous System Activities				Parasympathetic Nervous System Activity		
HRV Parameters	SDNN	HRV Index	Low Frequency	Total Power	rMSSD	pNN50	High Frequency
**Group characteristic**							
Age at the time of Fontan operation	**R = −0.379,** ***p*** **= 0.039**	R = −0.250, *p* = 0.183	R = −0.074, *p* = 0.697	R = −0.153, *p* = 0.421	R = 0.270, *p* = 0.149	**R = −0.422,** ***p*** **= 0.020**	R = −0.047, *p* = 0.805
Age during the study	**R = −0.426,** ***p*** **= 0.019**	R = −0.217, *p* = 0.25	R = −0.31, *p* = 0.094	**R = −0.432,** ***p*** **= 0.017**	R = −0.061, *p* = 0.749	**R = −0.481,** ***p*** **= 0.007**	R = −0.205, *p* = 0.276
**Echocardiography**							
Ejection fraction of the systemic ventricle	**R = 0.584,** ***p*** **= 0.001**	**R = 0.424,** ***p*** **= 0.02**	R = 0.347, *p* = 0.06	**R = 0.453,** ***p*** **= 0.012**	R = 0.032, *p* = 0.104	**R = 0.638,** ***p*** **= 0.001**	R = 0.281, *p* = 0.132
**Chronotropic parameters**							
HRR	R = −0.137, *p* = 0.471	**R = −0.411,** ***p*** **= 0.023**	R = −0.221, *p* = 0.240	R = −0.264, *p* = 0.159	R = −0.323, *p* = 0.082	R = −0.262, *p* = 0.162	R = −0.193, *p* = 0.307
Chronotropic index	R = 0.220, *p* = 0.243	R = 0.330, *p* = 0.075	R = 0.332, *p* = 0.073	R = 0.333, *p* = 0.072	R = 0.291, *p* = 0.118	**R = 0.419,** ***p*** **= 0.021**	R = 0.164, *p* = 0.386
**CPET parameters**							
Exercise time	R = 0.318, *p* = 0.086	**R = 0.709,** ***p* < 0.001**	**R = 0.544,** ***p*** **= 0.002**	**R = 0.45,** ***p*** **= 0.013**	R = −0.099, *p* = 0.601	**R = 0.440,** ***p*** **= 0.015**	**R = 0.38,** ***p*** **= 0.038**
Peak heart rate	R = 0.262, *p* = 0.163	**R = 0.485,** ***p*** **= 0.007**	**R = 0.384,** ***p*** **= 0.036**	**R = 0.375,** ***p*** **= 0.041**	**R = 0.377,** ***p*** **= 0.040**	**R = 0.394,** ***p*** **= 0.031**	R = 0.277, *p* = 0.138
Peak VO_2_ per kg	**R = 0.404,** ***p*** **= 0.027**	**R = 0.607,** ***p*** **< 0.001**	**R = 0.394,** ***p*** **= 0.031**	R = 0.360, *p* = 0.051	R = 0.142, *p* = 0.453	**R = 0.524,** ***p* = 0.003**	R = 0.290, *p* = 0.119
Peak VO_2_, %N	R = 0.137, *p* = 0.469	**R = 0.541,** ***p*** **= 0.002**	R = 0.233, *p* = 0.216	R = 0.185, *p* = 0.328	R = −0.016, *p* = 0.933	R = 0.222, *p* = 0.238	R = 0.077, *p* = 0.687
VE	**R = 0.434,** ***p*** **= 0.017**	**R = 0.369,** ***p*** **= 0.045**	R = 0.274, *p* = 0.143	R = 0.249, *p* = 0.184	R = 0.076, *p* = 0.690	R = 0.298, *p* = 0.110	**R = 0.377,** ***p* = 0.04**
VE/VCO_2_	R = −0.012, *p* = 0.951	**R = −0.424,** ***p*** **= 0.019**	R = −0.240, *p* = 0.202	R = −0.156, *p* = 0.411	R = 0.119, *p* = 0.530	R = −0.305, *p* = 0.102	R = −0.197, *p* = 0.296
**Laboratory tests**							
GGTP	R = −0.322, *p* = 0.083	R = −0.245, *p* = 0.192	**R = −0.385,** ***p* = 0.036**	R = −0.346, *p* = 0.061	R = −0.154, *p* = 0.415	**R = −0.368,** ***p*** **= 0.046**	R = −0.309, *p* = 0.096

Abbreviations: CPET, cardiopulmonary exercise test; GGTP, γ-glutamyl transpeptidase; HRR, heart rate reserve; HRV, heart rate variability; peak VO_2_ per kg, peak oxygen uptake per kilogram; peak VO_2_ (%N), percentage of predicted value for peak oxygen uptake; pNN50, percentage difference between adjacent NN intervals of >50 ms duration; rMSSD, root mean square of the differences of successive NN intervals; SDNN, standard deviation of all NN intervals; VE, minute ventilation; VE/VCO_2_, peak ventilatory equivalent of CO_2_. Significant results in bold.

**Table 7 jcm-10-04492-t007:** Association between oxygen saturation at rest, pNN50, peak heart rate, and chronotropic index.

	Univariable Analysis				Multivariable Analysis			
	*b*	95% CI	*p*-Value	R^2^	*b*	95% CI	*p*-Value	R^2^
Sat. O_2_ rest	0.026	(0.004–0.047)	0.02	0.18	0.021	(0.006–0.035)	0.006	
pNN50	0.016	(0.003–0.029)	0.018	0.19	0.011	(0.003–0.02)	0.013	0.7
Peak heart rate	0.005	(0.003–0.007)	<0.001	0.54	0.004	(0.002–0.006)	<0.001	

Abbreviations: HRR, heart rate reserve; pNN50, percentage difference between adjacent NN intervals of >50 ms duration; Sat. O_2_, oxygen saturation; 95% CI, 95% confidence interval; *b*—coefficient from linear regression; R^2^—coefficient of determination.

## Data Availability

The data presented in this study are available on request from the corresponding author. The data are not publicly available due to planned further publications.

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
