# Peer review of "Heart Rate Variability and Its Associations with Organ Complications in Adults after Fontan Operation"

_jcm, 2021, doi:10.3390/jcm10194492_

Round 1
Reviewer 1 Report
I would like to congratulate the Authors. The study raises an important issue, and has significant clinical implications.
This paper presents analysis and significance of heart rate variability (HRV) in patients after Fontan operation. Based on the literature HRV has been previously described as an independent risk factor for heart failure development, premature mortality etc.
This article covers the topic and highlights very important issue of decreased HRV in patients after FO and it’s prognostic relevance.
The authors have sufficiently listed the limitations of the study.
There are only several minor comments / suggestions to the manuscript:
- Please suggest a short recommendation for the use of the results in practice in prevention of adverse events after FO
- In the aim of the study you mentioned that “multiorgan complications” will be assessed. Could you please specify in the methods section this term and how it will be measured?
- have you performed an analysis of the correlation of echocardiographic parameters (e.g. EF) with HRV?
Reviewer 2 Report
Major
This study aimed to assess the association between HRV, exercise capacity, and multiorgan complications in adults after FO. They included 30 adult Fontan patients and 30 normal individuals.
1. However, in this study, there were very limited data about multiorgan complications.
Fontan circulation is characterized by increased central venous pressure from a lack of subpulmonic ventricle, low cardiac output, and chronic hypoxemia. This condition results in multiorgan complications; chronic liver disease, such as liver fibrosis/cirrhosis, chronic kidney disease, protein losing enteropathy, and so on. However, in this study, only gamma-glutamyl GT was described as multiorgan complications. So they need to add the data for the multiorgan complications.
For example, Liver complications include elevation of bilirubin, thrombocytopenia, liver nodules, and liver cirrhosis on sono or CT image.. as well as GGT elevations.
- There are scarce data about baseline characteristics for patients in this study. Authors should consider to include age at Fontan surgery, follow-up time after Fontan, Fontan type (lateral tunnel or extracardiac conduit Fontan), systemic ventricle type (RV, LV, or both?), presence of heterotaxia, systemic ventricular function, valvular abnormality, valvular abnormality, BNP, catheterization data if available, NYHA functional class, and so on.
- In table 3, RER peak was 1.0 (SD 0.08) in the patient group. However, An RER > 1.05-1.09 is considered to be compatible with a good effort. Peak RER <1.00 generally reflects submaximal cardiovascular effort and caution should be taken in the use of peak VO2 for prognostic purposes in the presence of a low peak RER. Is the peak VO2 and other parameres of cardiopulmonary exercise test reliable in patients with RER<1.0? Can you explain this?
Mino
- They excluded the patients with atrial flutter and atrial fibrillation. Is it means atrial flutter and AF during Holter monitoring?
- Were there any difference in HRV between Fontan type, or heterotaxy syndrome
- There are too many abbreviations, around 24.
Ex. SVEF was used only 3 times in the manuscript.
The authors need to avoid the use of the unnecessary abbreviation
4. In table 2, which type of total cavopulmonary connection was performed? Lateral tunnel type or extracardiac conduit Fontan type?
5. In line 138, what does TP mean?
6. In Tables 4,5, and 6, there is no description for the abbreviations used in the table.
8. In line 192, HRR peak -> HR peak
